# A Novel Squalenoylated Temozolomide Nanoparticle with Long Circulating Properties Reverses Drug Resistance in Glioblastoma

**DOI:** 10.3390/ijms26104723

**Published:** 2025-05-15

**Authors:** Jiao Feng, Chengyong Wen, Xiao Zhang, Xiaolong Zhu, Mengmeng Ma, Xiaohong Zhao, Xinbing Sui

**Affiliations:** 1School of Pharmacy, Hangzhou Normal University, Hangzhou 311121, China; wency02@163.com (C.W.);; 2Shuren College, Zhejiang Chinese Medical University, Hangzhou 310053, China

**Keywords:** glioblastoma, temozolomide resistance, squalenoylation, MGMT expression

## Abstract

Temozolomide (TMZ) remains the frontline chemotherapy for gliomas; yet its clinical efficacy is significantly compromised by inherent instability and the emergence of resistance mechanisms. To surmount these challenges, we engineered a squalenoylated TMZ nanoparticle (SQ-TMZ NPs) via conjugation of TMZ with squalene, enabling enhanced drug stability and improved therapeutic potency against glioblastoma cells. The resulting SQ-TMZ NPs exhibited a precisely controlled nanoscale architecture (~126 nm), demonstrating exceptional stability under physiological and storage conditions, with minimal hemolytic toxicity (<5%). Notably, these nanoparticles conferred superior cytotoxicity in TMZ-resistant glioblastoma T98G cells, attributed to the amplification of intracellular reactive oxygen species (ROS) and DNA damage, along with MGMT (O-6-methylguanine-DNA methyltransferase) expression suppression. Furthermore, in vivo imaging confirmed their efficient blood–brain barrier (BBB) penetration and selective tumor accumulation. This study presents a transformative approach by integrating prodrug self-assembly with targeted drug delivery to not only enhance TMZ stability but also decisively reverse glioblastoma resistance, offering a compelling therapeutic advancement.

## 1. Introduction

Glioblastoma multiforme (GBM) is the most aggressive and malignant form of primary brain tumor in the brain, accounting for 49.1% of all malignant brain tumors [1]. Characterized by rapid growth, high invasiveness, and intrinsic resistance to conventional therapies, GBM remains a formidable challenge in oncology [2]. The standard treatment regimen for GBM includes surgical resection, radiation therapy, and chemotherapy, typically with the chemotherapeutic agent TMZ [3]. However, the development of TMZ resistance, both intrinsic and acquired, has emerged as a significant challenge, limiting its therapeutic efficacy in GBM management. This resistance, coupled with the tumor’s rapid progression, results in a poor prognosis for patients, with the five-year survival rate remaining critically low [4]. The prognosis for GBM remains extremely poor, with a median overall survival of only 15 months and a five-year survival rate of less than 5% [5].

The primary mechanisms of TMZ resistance include upregulation of DNA repair systems such as the mismatch repair (MMR) [6] and base excision repair (BER) pathways, as well as elevated expression of O^6^-methylguanine-DNA methyltransferase (MGMT) [7,8]. Among these, MGMT directly counteracts TMZ’s alkylating action by removing methyl adducts from the O^6^ position of guanine, thereby preventing DNA damage and facilitating tumor survival [9]. High MGMT expression has been strongly correlated with poor TMZ response, underscoring the urgent need to overcome this resistance mechanism [10]. Beyond resistance, TMZ also has inherent limitations, including a short half-life.

To address these limitations, numerous drug delivery strategies and chemical modifications have been explored to enhance TMZ stability, prolong circulation time, and improve tumor targeting. Among these strategies, lipid-based delivery systems have gained considerable attention [11,12]. Squalene, a naturally occurring triterpene, has demonstrated promising potential as a biocompatible carrier capable of self-assembly into nanoparticles [13,14]. By leveraging the self-assembling properties of TMZ, we developed squalenoylated TMZ nanoparticles (SQ-TMZ NPs) to enhance TMZ stability, prolong systemic circulation, and improve its therapeutic efficacy against resistant glioblastoma cells.

## 2. Results

### 2.1. Synthesis of Squalene-Temozolomide Conjugate

The SQ-TMZ conjugate was synthesized via a synthetic route depicted in Figure 1. Briefly, squalene was first chemically modified at one terminus to introduce a hydroxyl group [15], which was subsequently conjugated to TMZ via an ester linkage under the catalysis of EDC and DMAP. The final waxy polymeric conjugate was obtained in 60.39% yield after purification by silica gel column chromatography. Structural confirmation of the target compound was performed using NMR and HR-MS, corroborating the successful formation of the SQ-TMZ conjugate (Appendix A).

### 2.2. Characterization and Stability of SQ-TMZ NPs

SQ-TMZ NPs were successfully prepared using a nanoprecipitation method, resulting in monodisperse nanoparticles with an average hydrodynamic diameter of ~126 nm and a low polydispersity index (PDI), indicative of uniform size distribution (Figure 2A–C, Appendix A). TMZ exhibits a characteristic absorption peak at 328 nm. UV–Vis spectroscopy demonstrated that SQ-TMZ retained the characteristic absorption peaks of both squalene and TMZ, indicating structural stability and preserved pharmacophore integrity (Figure 2D and Appendix A). Building on this, the colloidal and chemical stability of the nanoparticles was further investigated over a 7-day storage period at 4 °C. Dynamic light scattering (DLS) measurements showed minimal variation in particle size (<10 nm increase) and consistently low PDI values, indicating that the nanoparticles remained physically stable (Figure 2E). Simultaneously, UV–Vis spectra showed no significant decrease in the 328 nm absorption peak, suggesting that TMZ within the nanoparticles remained chemically intact and that the nanoparticle matrix effectively protected it from degradation (Figure 2F). Together, these results demonstrate that the SQ-TMZ NPs possess excellent colloidal and chemical stability, making them promising candidates for further development.

We further investigated the stability of the NPs under different pH conditions as well as in in vitro/in vivo environments. TMZ is highly unstable under alkaline conditions and prone to degradation. The UV–Vis absorption spectra of free TMZ and SQ-TMZ NPs were analyzed in phosphate buffer solutions at pH 5.0, 6.4, and 7.4. Free TMZ was found to be highly unstable under alkaline conditions and prone to rapid degradation across all tested pH levels. In contrast, SQ-TMZ NPs remained structurally intact under both mildly acidic and mildly basic conditions (pH 5.0–7.4), demonstrating significantly enhanced stability (Figure 3A–F,K). This sustained structural integrity under physiological pH conditions indicates the potential of SQ-TMZ NPs for prolonged circulation and improved bioavailability.

To assess drug stability in biological environments, we further examined the behavior of SQ-TMZ NPs and free TMZ in mouse serum and DMEM cell culture medium. UV–Vis absorption spectra revealed that both formulations underwent degradation within 12 h. However, SQ-TMZ NPs showed a more gradual degradation profile. Upon normalizing the absorbance at 328 nm, it was observed that free TMZ degraded rapidly, with almost complete conversion to MTIC within 6 h. In contrast, a substantial amount of intact TMZ remained in the SQ-TMZ NPs even after 12 h, indicating a slower degradation rate (Figure 3G–J,L). These results suggest that SQ-TMZ NPs possess improved stability in biological media, potentially contributing to a prolonged therapeutic effect.

### 2.3. Cytotoxicity and Resistance Reversal in GBM Cells

The high expression of MGMT in tumor cells reduces their sensitivity to alkylating agents, leading to drug resistance. Based on this, MGMT expression is commonly used as an indicator of TMZ resistance [9]. According to the data from the HPA databases, MGMT is highly expressed in 50% of brain tumor cell lines (Appendix A). Guided by this information, three cell lines were selected for the experiment: T98G (high MGMT expression) and U87 and U251 (low MGMT expression). Western blotting results confirmed that MGMT was highly expressed in T98G cells, while U87 and U251 showed negligible expression, consistent with the database results (Figure 4A). Subsequent cytotoxicity assays using CCK-8 demonstrated that T98G cells were significantly more resistant to TMZ treatment compared to U87 and U251 (Appendix A). Specifically, the IC_50_ value for T98G cells was 5.78 mM, whereas U87 and U251 cells had IC_50_ values of 1.51 mM and 2.88 mM, respectively (Figure 4B). These results further validate that T98G is a TMZ-resistant cell line, while U87 and U251 are TMZ-sensitive.

Based on the previous findings, T98G was selected as the TMZ-resistant cell line, while U87 and U251 were used as TMZ-sensitive cell lines for subsequent studies. The antitumor effects of SQ-TMZ NPs were then evaluated in these cell lines at multiple levels, including cytotoxicity, colony formation, and apoptosis induction. Since SQ-TMZ is chemically conjugated, we also included a control group consisting of a physical mixture of TMZ and SQ NPs to investigate whether this combination could produce comparable effects to the SQ-TMZ conjugate. First, cytotoxicity assays revealed that SQ-TMZ NPs exhibited significantly stronger cytotoxic effects than free TMZ in both resistant and sensitive cell lines (Appendix A). Although SQ NPs alone showed some cytotoxicity at high concentrations, the SQ-TMZ prodrug—formed by chemical conjugation of TMZ and SQ—was more effective than a physical mixture of the two components. This suggests that the enhanced effect of SQ-TMZ NPs is not due to simple drug combination but may involve a novel mechanism of action. Notably, the IC_50_ values for TMZ were 537, 41, and 115 times higher than those of SQ-TMZ NPs in T98G, U87, and U251 cells, respectively, indicating that chemical modification and nanoassembly of TMZ significantly improve its antitumor efficacy, especially in resistant cells (Figure 4B–E). This approach may offer a new strategy to overcome TMZ resistance. In colony formation assays, TMZ treatment did not significantly inhibit colony formation in T98G cells, reaffirming their resistance. Specifically, at a concentration of 10 μM, TMZ showed no observable effect on the colony-forming ability of T98G cells compared to the control group. In contrast, treatment with SQ-TMZ NPs at the same concentration resulted in a substantial reduction in colony area, indicating a significantly stronger ability to suppress long-term cell proliferation. In the TMZ-sensitive U87 and U251 cells, TMZ at 40 μM did reduce colony formation, demonstrating its efficacy in these cell lines. However, SQ-TMZ NPs at the same concentration led to even more pronounced inhibition, suggesting that the nanoformulated prodrug has superior long-term antiproliferative effects even in sensitive glioma cells (Figure 4F–H). These results highlight the potential of SQ-TMZ NPs to enhance therapeutic outcomes across different glioma cell types.

The Annexin V/PI results indicate that at the same concentration, the apoptosis rate in the SQ-TMZ NPs group was significantly higher than in the free TMZ group. In T98G and U251 cells, early apoptotic cells were significantly more prevalent than late apoptotic cells. These results suggest that SQ-TMZ NPs have a significantly stronger ability to induce apoptosis compared to free TMZ, which is consistent with the colony formation assay results (Figure 4I–K).

### 2.4. SQ-TMZ NPs Induce ROS Accumulation and Enhance DNA Damage to Overcome TMZ Resistance in GBM Cells

To investigate the mechanism by which SQ-TMZ NPs overcome TMZ resistance, intracellular reactive oxygen species (ROS) levels were assessed using the DCFH-DA probe after 48 h of drug treatment. As shown in Figure 5A, both the TMZ and SQ NPs groups exhibited very weak green fluorescence, comparable to the control group. In contrast, cells treated with SQ-TMZ NPs displayed a markedly stronger green fluorescence signal, indicating elevated ROS levels. Notably, the ROS signal in the SQ-TMZ NP group was significantly higher than that induced by free TMZ.

Flow cytometry analysis further quantified this effect, showing that SQ-TMZ NPs triggered substantial ROS accumulation in both TMZ-sensitive U87 and TMZ-resistant T98G cells. In U87 cells, ROS levels reached 80.0%, whereas in T98G cells, ROS accumulation was lower at 21.1%, yet still significantly higher than in the control and TMZ groups. These results demonstrate that SQ-TMZ NPs effectively induce ROS in both cell lines, although ROS may not be the dominant mechanism of cytotoxicity in T98G cells (Figure 5B).

To evaluate DNA damage, we examined γ-H_2_AX expression, a biomarker of DNA double-strand breaks. Immunofluorescence and Western blot analyses showed that TMZ treatment resulted in γ-H_2_AX expression in both U87 and T98G cells. In contrast, SQ-TMZ NP treatment led to intense γ-H_2_AX fluorescence in both cell lines, surpassing levels observed in all other groups, including free TMZ and SQ NPs. Western blot results confirmed these observations, revealing significantly elevated γ-H_2_AX protein levels and enhanced histone γ-H_2_AX phosphorylation following SQ-TMZ NP treatment. Interestingly, SQ NPs alone also induced moderate DNA damage in T98G cells (Figure 6A–D).

Since MGMT expression is a key factor in TMZ resistance, we further explored the mechanism by examining the impact of SQ-TMZ NPs on MGMT expression in T98G cells using RT-PCR and Western blotting. The results revealed that, at the transcriptional level, SQ-TMZ NPs dose-dependently suppressed MGMT gene expression more effectively than free TMZ. At the protein level, SQ-TMZ NPs significantly downregulated MGMT expression and almost completely inhibited MGMT protein production (Figure 6E–G). These findings suggest that SQ-TMZ NPs enhance the sensitivity of resistant T98G cells to TMZ by effectively suppressing MGMT expression, providing further evidence of their potential as a promising therapeutic strategy for overcoming TMZ resistance in glioma cells (Figure 6E–G).

### 2.5. SQ-TMZ NPs Exhibit Favorable Blood Compatibility and Efficient Blood–Brain Barrier Penetration

To assess the biosafety of SQ-TMZ NPs, a hemolysis assay was conducted. As shown in Figure 7A,B, both SQ-TMZ NPs and SQ NPs caused negligible hemolysis across different concentrations. Although a slight increase in hemolysis rate was observed at higher concentrations, all values remained below 5%, which is within the acceptable threshold for biomedical applications. These findings confirm that SQ-TMZ NPs have excellent blood compatibility and are safe for intravenous administration.

The ability of SQ-TMZ NPs to cross the blood–brain barrier (BBB) was further evaluated using DIR-labeled nanoparticles and in vivo fluorescence imaging. Following intravenous injection via the tail vein, strong fluorescence signals were observed in the brain within 2 h, indicating successful BBB penetration. Mice treated with SQ-TMZ NPs showed significant fluorescence accumulation in the brain region, while those injected with free DIR dye exhibited fluorescence primarily in the liver, with no signal detected in the brain. This demonstrates the efficient brain-targeting capability of the SQ-TMZ NPs.

Notably, fluorescence intensity in the brain continued to rise over time, reaching 1.71 × 10^7^ ph/s/cm^2^/sr at 24 h post-injection. The most pronounced increase occurred between 2 and 8 h, suggesting rapid and efficient nanoparticle transport across the BBB, along with sustained retention in brain tissue. These results indicate that SQ-TMZ NPs not only evade rapid clearance but also ensure targeted drug delivery to brain tumors, supporting their potential for effective glioma therapy (Figure 7C,D).

## 3. Discussion

Temozolomide (TMZ) is the current standard chemotherapy for glioblastoma (GBM), yet its therapeutic efficacy is severely hampered by poor chemical stability, and frequent development of resistance, particularly through MGMT upregulation [16]. To overcome these challenges, nanoparticle-based drug delivery systems have been explored to enhance TMZ’s stability, bioavailability, and tumor targeting [17]. Among these, polymer–drug conjugate nanocarriers offer distinct advantages over traditional nanoparticles. By covalently linking a drug to a polymer backbone, they enable nearly 100% drug loading and minimize the burst release commonly associated with physical encapsulation methods. These conjugates can spontaneously self-assemble into nanoparticles without the need for complex emulsification or solvent evaporation processes. Moreover, they exhibit enhanced stability under physiological and storage conditions, making them a highly promising platform for more effective and controlled TMZ delivery [18].

Notably, Du et al. introduced a dual-stimuli-responsive nanoparticle (P1SNP) that co-delivers MTIC, the active TMZ metabolite, and nitric oxide (NO) to boost cytotoxicity [19], while Sharma et al. synthesized a novel Naphthalimide-Selenourea-TMZ conjugate (Naph-Se-TMZ) that combines DNA intercalation, ROS induction, and HDAC1 inhibition [20]. Our study presents a squalenoylated TMZ nanoparticle formulation that not only significantly enhances the stability of TMZ but also effectively reverses drug resistance in glioblastoma cells by downregulating MTMG expression.

One of the most pressing limitations of conventional TMZ therapy is its poor chemical stability, especially in physiological conditions. Free TMZ is rapidly hydrolyzed to its active metabolite, 5-(3-methyltriazen-1-yl) imidazole-4-carboxamide (MTIC), leading to a short half-life and reduced bioavailability [21]. Our results demonstrate that SQ-TMZ NPs exhibit superior stability compared to free TMZ, as evidenced by their sustained structural integrity under a range of pH conditions (pH 5.0–7.4) and prolonged degradation kinetics in biological fluids. This enhanced stability likely contributes to prolonged systemic circulation and improved therapeutic efficacy in vivo.

In line with previous findings that MGMT is a major determinant of TMZ resistance in glioblastoma, our study shows that SQ-TMZ nanoparticles significantly downregulate MGMT expression in TMZ-resistant T98G cells, thereby resensitizing them to alkylating therapy. Although the precise mechanism of MGMT suppression by SQ-TMZ NPs requires further elucidation, Wang et al. developed an siRNA-loaded nanoparticle system that effectively silenced MGMT in orthotopic GBM models, leading to enhanced TMZ sensitivity and a substantial survival benefit in vivo [22]. These results further reinforce that nanoparticle-mediated MGMT suppression—whether through genetic (siRNA) or chemical (drug-induced) pathways—represents a promising strategy to overcome TMZ resistance in GBM therapy.

Furthermore, our in vivo imaging confirmed that SQ-TMZ NPs can effectively cross the blood–brain barrier (BBB) and preferentially accumulate in brain tumor tissue, an essential requirement for GBM therapy. While several BBB-penetrating nanocarriers have been reported, such as receptor-targeted peptides [23], our results demonstrate that passive targeting via the EPR effect and the lipid-based nature of the formulation can achieve efficient brain delivery without additional surface modifications.

While the in vitro results are promising, a key limitation of this study is the absence of in vivo efficacy evaluation in a GBM orthotopic model. Although biodistribution was assessed, future work should include tumor growth inhibition and survival analysis to confirm therapeutic benefit. Similarly, mechanistic exploration of MGMT suppression through gain- and loss-of-function experiments will provide deeper insights into the molecular basis of resistance reversal.

In summary, SQ-TMZ NPs represent a promising TMZ delivery platform that enhances stability, promotes BBB penetration, and overcomes drug resistance via MGMT suppression and oxidative stress induction. This strategy complements existing delivery technologies and offers a simplified, biocompatible alternative with translational potential, warranting further preclinical investigation.

## 4. Materials and Methods

### 4.1. Materials

Squalene and sodium borohydride (NaBH_4_) were purchased from Aladdin (Shanghai, China). N-Bromosuccinimide (NBS), K_2_CO_3_, Periodic acid (HIO_4_·2H_2_O), ethylene dichloride (EDC), 4-dimethylaminopyridine (DMAP), and TMZ acid were purchased from Macklin (Shanghai, China). Opti-MEM cell culture medium was purchased from Gibco. DMEM cell culture medium, penicillin-streptomycin solution, and trypsin were from Basalmedia (Shanghai, China). RIPA lysis buffer, PMSF, SDS-PAGE sample loading buffer, and DAPI were from Beyotime (Shanghai, China). Annexin V-FITC apoptosis detection kit was from Yeasen (Shanghai, China). Anti-MGMT, Anti-γ-H_2_AX, Anti-GAPDH, Anti-rabbit lgG, and Anti-mouse lgG were obtained from Cell Signaling Technology, Inc. (Danvers, MA, USA).

### 4.2. Synthesis of Squalene-Temozolomide (SQ-TMZ) Copolymer

1,1′,2-Tris-norsqualenol was synthesized from squalene via 1,1′,2-tris-norsqualenic aldehyde according to previously reported methods [15]. TMZ-carboxylic acid (40 mg, 0.2 mmol) was suspended in DCM (5 mL). 1,1′,2-Tris-norsqualenol and (79 mg, 0.2 mmol) and catalytic DMAP were added to the suspension (2.5 mg, 0.02 mmol), followed by EDC (47.2 mg, 0.24 mmol). After stirring at room temperature under argon for 14 h, the mixture was filtered, extracted by DCM (10 mL), and washed with aqueous 0.1 M HCl. The organic phase was separated, dried over Na_2_SO_4_, and concentrated under reduced pressure. The product was purified by silica column chromatography (PE/EA = 7.5:2.5), obtained in 60.39% yield as a white solid. ^1^HNMR (400 MHz, CDCl_3_) δ, ppm: 8.44 (s, 1H), 5.21–5.02 (m, 5H), 4.42 (t, J = 6.8 Hz, 2H), 4.03 (s, 3H), 2.13 (t, J = 7.4 Hz, 2H), 1.98 (tt, J = 18.2, 7.1 Hz, 18H), 1.65 (s, 3H), 1.59 (d, J = 13.7 Hz, 15H). ^13^C NMR (101 MHz, CDCl3) δ 160.43 (s), 138.59 (s), 135.73 (s), 135.24–134.79 (m), 133.49 (s), 131.24 (s), 129.45 (s), 128.52 (s), 125.28 (s), 124.32 (dd, J = 11.2, 3.3 Hz), 65.60 (s), 39.88–39.46 (m), 36.63 (s), 35.65 (s), 28.25 (s), 26.71 (dd, J = 10.7, 2.0 Hz), 25.70 (s), 17.68 (s), 16.14–15.70 (m). HR-MS(*m*/*z*): Calcd. for C_33_H_50_N_5_NaO_3_ [M + Na]^+^: 586.3749. Found: [M + Na]^+^ = 586.3728.

### 4.3. Preparation of SQ-TMZ Nanoparticles (SQ-TMZ NPs)

SQ-TMZ NPs were prepared using the nanoprecipitation method. Briefly, SQ-TMZ conjugate and DSPE-mPEG2000 were dissolved in a mixture of EtOH and THF (*V*_EtOH_: *V*_THF_ = 2:1). The organic solution was added dropwise under stirring (500× *g* rpm) into a 5% aqueous dextrose solution (volume ratio EtOH/THF: dextrose solution = 1:4). The solution became spontaneously turbid with a Tyndall effect, indicating the formation of the nanoparticles. Ethanol and THF were then completely evaporated using a rotavapor (80× *g* rpm, 30 °C, 30 mbar) to obtain an aqueous suspension of pure SQ-TMZ NPs. Blank SQ NPs (TMZ-free NPs) were prepared by the same method as described above by adding dropwise an organic solution of squalenol into the aqueous solution.

### 4.4. Characterization of SQ-TMZ Nanoparticles (SQ-TMZ NPs)

The mean particle size, polydispersity index (PDI), and zeta potential were primarily evaluated by DLS (Nano ZS, Malvern, Worcestershire, UK). The morphology of the nanoparticles was observed using transmission electron microscopy (TEM) (Hitachi, Tokyo, Japan) and cryo-transmission electron microscopy (cryo-TEM) (Thermo Fisher, Waltham, MA, USA).

### 4.5. Cytotoxicity of SQ-TMZ NPs

The cytotoxicity of SQ-TMZ NPs on human GBM cells lines (U87, U251 and T98G) was determined using CCK-8 (Meilunbio, Dalian, China, Cat.: MA0218). Briefly, U87, U251, or T98G cells were seeded in 96-well plates (3 × 10^3^ cells/well) and incubated with free TMZ, SQ-TMZ NPs, and SQ NPs. After 48 h incubation, 100 μL DMEM medium including 10% CCK-8 was added for each well, followed by incubation at 37 °C for 1–4 h, then measured at 450 nm wavelength.

### 4.6. Colony Formation Assay

To evaluate the effect of different concentrations of SQ-TMZ NPs on glioma cell colony formation, a colony formation assay was performed as follows: A total of 5000 cells were seeded into 6 cm culture dishes and incubated under standard culture conditions, ensuring uniform dispersion without cell aggregation. Once cells adhered, microscopic observation confirmed that they were in a single-cell state. TMZ, SQ NPs, and SQ-TMZ NPs solutions of different concentrations were added to each dish. The culture was continued for 10–14 days until visible colonies formed. When 40–50 cell colonies were observed under a microscope, the cells were gently washed three times with PBS. Colonies were then fixed with polyformaldehyde for 20 min and washed again three times with PBS. Then, 3 mL of crystal violet staining solution was added, and plates were incubated at room temperature for 30 min. After staining, the crystal violet solution was removed, and the plates were gently washed with distilled water until the rinse water became colorless and transparent. The plates were then tilted and placed upside down on absorbent paper to air dry. Finally, images of the stained colonies were captured for analysis.

### 4.7. Apoptosis of SQ-TMZ NPs

To evaluate the therapeutic efficacy of SQ-TMZ NPs, apoptosis was assessed using flow cytometry following the protocol provided by the Yeasen apoptosis detection kit. The specific steps were as follows: Cells were seeded into 6-well plates at a density of 1.5–3.0 × 10^5^ cells/mL per well and incubated overnight. Once cells adhered completely, TMZ, SQ NPs, and SQ-TMZ NPs solutions of different concentrations were added. The cells were incubated at 37 °C with 5% CO_2_ for 48 h. After incubation, cells were collected and stained with Annexin V-FITC and PI dyes and incubated on ice in the dark for 20 min. The samples were analyzed by flow cytometer to determine the proportion of apoptotic cells.

### 4.8. Western Blot

Wells were harvested, washed with PBS, and lysed with RIPA buffer (100:1:1, RIPA: PMSF: phosphatase inhibitor). Lysates were sonicated (30 s on/30 s off, 20 min) and centrifuged (12,000× *g* rpm, 20 min, 4 °C) to collect the supernatant. Protein concentration was determined using BCA assay, and samples were adjusted to equal concentrations, mixed with 5× SDS loading buffer, heated at 100 °C for 10 min, and stored at −80 °C. Proteins were separated by SDS-PAGE (80 V stacking, 120–150 V separation) and transferred onto PVDF membranes (250 mA, 1.5–2 h, 4 °C). Membranes were blocked with 5% milk, incubated with primary antibodies (1:1000, overnight, 4 °C) and secondary antibodies (1–2 h, room temperature). Signals were detected using ECL reagent and imaged using chemiluminescence imaging system (Bio-Rad, Hercules, CA, USA).

### 4.9. Quantitative RT-PCR Analysis of MGMT mRNA Expression

T98G cells were seeded at a density of 5.0 × 10^6^ cells per well in 6 cm dishes and treated with free TMZ, SQ NPs, or SQ-TMZ NPs for 24 h. Total RNA was extracted using Trizol reagent, followed by phase separation with chloroform and RNA precipitation with isopropanol. The RNA pellet was washed with 75% ethanol, air-dried, and dissolved in RNase-free water. RNA concentration and purity were measured before reverse transcription. cDNA synthesis was performed at 37 °C for 15 min, followed by 85 °C for 5 s. qPCR was conducted using ChamQ Universal SYBR qPCR Master Mix, with MGMT and GAPDH primers. The qPCR cycling conditions were 95 °C for 5 min, followed by 40 cycles of 95 °C for 5 s, 60 °C for 30 s, and 72 °C for 30 s. Relative MGMT expression was determined using the Ct method. Primer information: forward MGMT primer: ACCGTTTGCGACTTGGTACTT; reverse MGMT primer: GGAGCTTTATTTCGTGCAGACC. Forward GAPDH primer: GGAGCGAGATCCCTCCAAAAT; reverse GAPDH primer: GGCTGTTGTCATACTTCTCATGG.

### 4.10. Immunofluorescence Staining

Immunofluorescence (IF) staining is based on the principle of specific binding between the target protein and a primary antibody, followed by the binding of a fluorescence-labeled secondary antibody to the primary antibody. The detailed protocol is as follows: U87 and T98G glioma cells in good condition were seeded at a density of 1.0 × 10^5^ cells per well. Once the cells adhered completely, cells were treated with free TMZ, SQ NPs, or SQ-TMZ NPs and incubated for 36 h. After incubation, cells were washed and fixed for 20 min at room temperature. Cells were then blocked at room temperature for 2 h to prevent nonspecific binding. The primary antibody γ-H2AX was diluted at 1:400 (antibody–dilution buffer, *v*/*v*) and incubated at 4 °C overnight. After incubation, cells were washed three times with PBS and incubated with secondary antibody at room temperature in the dark for 2 h. After washing three times with PBS after incubation, the cell nuclear was stained with 1 mL of 1× DAPI solution at room temperature in the dark for 15 min. Cells were then imaged under a fluorescence microscope to assess protein expression and localization.

### 4.11. Analysis of Intracellular ROS Levels

ROS can be generated through various mechanisms, and their accumulation in tumor cells can induce multiple forms of cell death. In this experiment, the intracellular ROS levels were detected using the Beyotime ROS assay kit and analyzed by both fluorescence microscopy and flow cytometry. The specific steps were as follows: U87 and T98G glioma cells were harvested and seeded into 6 cm culture dishes at a density of 1.5 × 10^5^ cells per well. Cells were treated with free TMZ, SQ NPs, or SQ-TMZ NPs, while the control group received no drug treatment. The plates were incubated under standard culture conditions for 48 h. After 48 h of incubation, the cells were incubated with DCFH-DA working solution at 37 °C for 30 min. After incubation, the plates were placed under a fluorescence microscope, and images were captured to observe ROS fluorescence intensity. Following imaging, the cells were collected, and intracellular ROS levels were quantified by flow cytometry analysis.

### 4.12. Brain Targeting of SQ-TMZ NPs

U87 cells were harvested, centrifuged, and resuspended at 2 × 10^8^ cells/mL. Male BALB/c nude mice were anesthetized and secured on a stereotaxic apparatus. After scalp disinfection and incision, a burr hole was drilled 1.8 mm right, 0.6 mm anterior to the bregma. Then, 3 μL of U87 cell suspension was injected 3.5 mm deep, followed by a 5-min pause before needle withdrawal. Seven days after tumor establishment, mice were divided into free DIR and SQ-TMZ NPs groups (*n* = 3). SQ-TMZ NPs were injected via the tail vein, and in vivo imaging was performed at 0, 2, 4, 8, 12, and 24 h. After euthanasia, major organs were excised for ex vivo imaging to analyze SQ-TMZ NPs distribution.

## 5. Conclusions

In summary, our study provides compelling evidence that SQ-TMZ NPs offer a significant therapeutic advantage over free TMZ by improving drug stability, enhancing BBB penetration, and overcoming MGMT-mediated resistance in glioblastoma cells. The dual mechanisms of MGMT suppression and ROS-induced cytotoxicity further strengthen the therapeutic potential of this novel nanoparticle formulation.

## 6. Patents

This research has been granted a Chinese patent, with the patent number ZL202211580588.3.

## Figures and Tables

**Figure 1 ijms-26-04723-f001:**
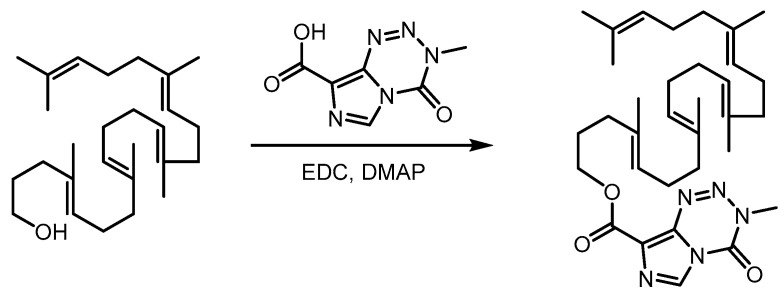
Synthesis of squalene-temozolomide (SQ-TMZ) copolymer.

**Figure 2 ijms-26-04723-f002:**
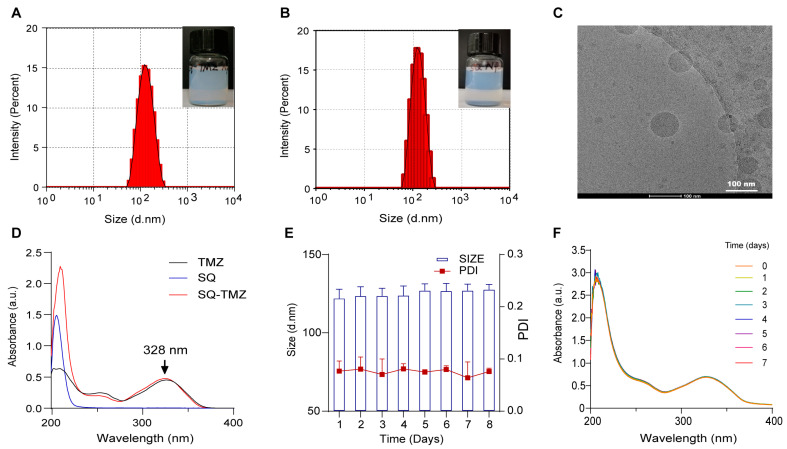
Characterization of SQ-TMZ NPs and their storage stability. (**A**,**B**) Particle size of SQ-TMZ NPs and SQ NPs; (**C**) Cryo-electron microscopy (cryo-EM) image of SQ-TMZ NPs; (**D**) UV–Vis spectra of TMZ, SQ, and SQ-TMZ; (**E**,**F**) Stability of SQ-TMZ NPs at 4 °C, evaluated by particle size (*n* = 3) (**E**) and UV–Vis absorption (**F**).

**Figure 3 ijms-26-04723-f003:**
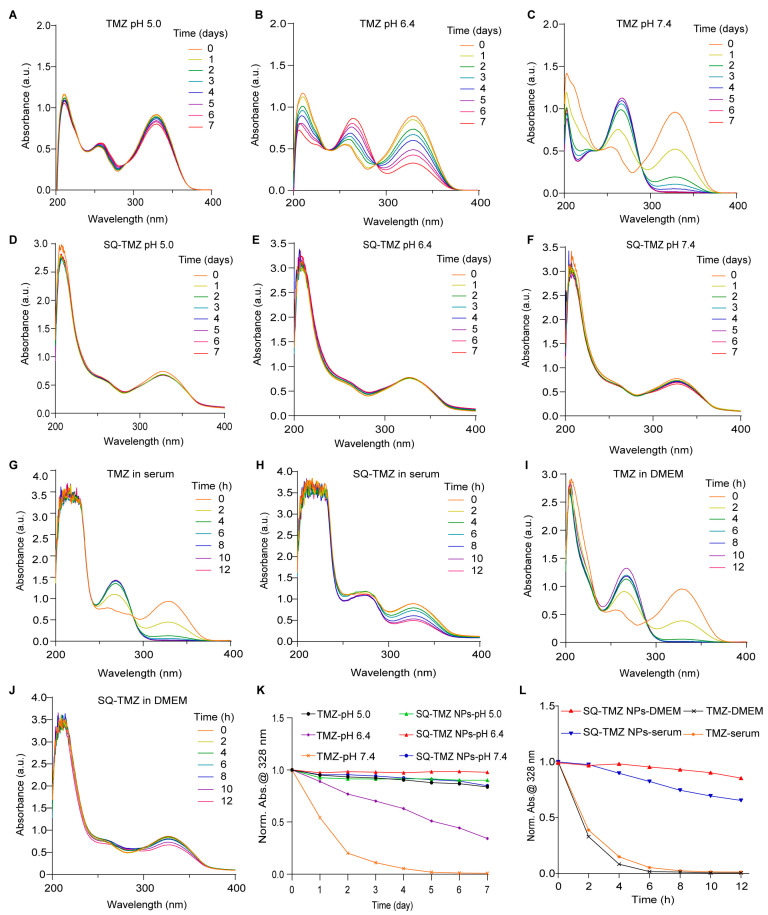
Enhanced stability of SQ-TMZ nanoparticles under physiological and biological conditions. (**A**–**C**) Stability of TMZ at pH 5.0, 6.4, and 7.4, measured by UV–Vis absorption; (**D**–**F**) Stability of SQ-TMZ NPs at pH 5.0, 6.4, and 7.4; (**G**,**H**) Comparison of the stability of TMZ and SQ-TMZ NPs in mouse serum; (**I**,**J**) Comparison of the stability of TMZ and SQ-TMZ NPs in DMEM medium; (**K**) Standardized UV–Vis absorbance analysis at 328 nm for free TMZ and SQ-TMZ NPs under varying pH conditions; (**L**) Standardized UV–Vis absorbance analysis at 328 nm for free TMZ and SQ-TMZ NPs in serum and DMEM medium.

**Figure 4 ijms-26-04723-f004:**
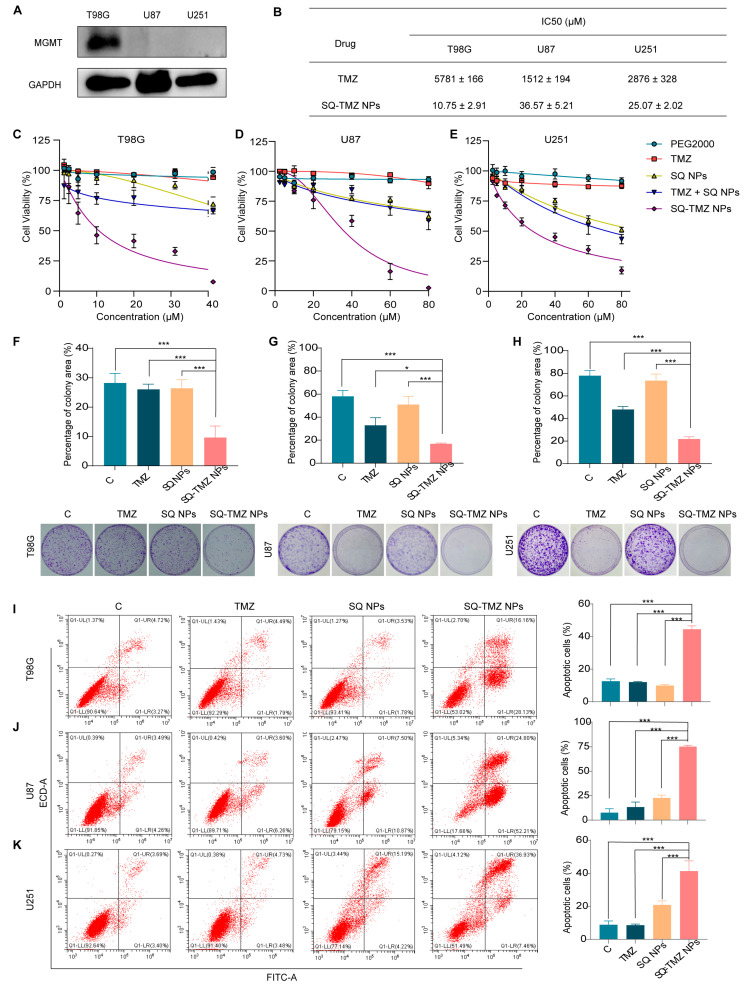
Cytotoxicity of SQ-TMZ NPs in TMZ-resistant and TMZ-sensitive cells. (**A**) MGMT protein expression levels in T98G, U251, and U87 cell lines. (**B**) IC_50_ values of TMZ in the three cell lines. (**C**–**E**) Cytotoxicity of TMZ, SQ NPs, and SQ-TMZ NPs in the three glioma cell lines, as determined by the CCK-8 assay. (**F**–**H**) Effects of TMZ, SQ NPs, and SQ-TMZ NPs on the proliferative ability of the three glioma cell lines, as assessed by a clonogenic assay. The drug concentration was 10 μM for T98G cells and 40 μM for U87 and U251 cells. (**I**–**K**) Effects of TMZ, SQ NPs, and SQ-TMZ NPs on apoptosis in the three glioma cell lines, as determined by flow cytometry. The drug concentration was 10 μM for T98G cells and 40 μM for U87 and U251 cells (mean ± SD, *n* = 3, One-way ANOVA was performed to compare group means, and post hoc multiple comparisons were conducted using Tukey’s HSD test, * vs. SQ-TMZ NPs, *** *p* < 0.001, * *p* < 0.05).

**Figure 5 ijms-26-04723-f005:**
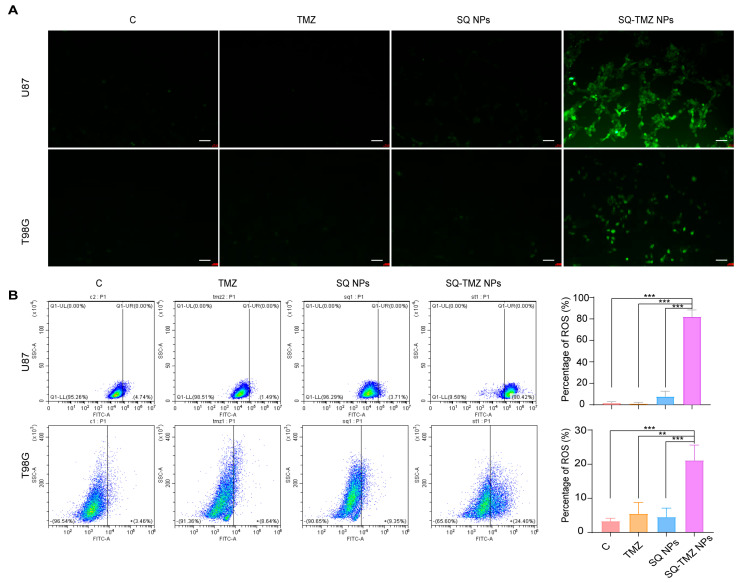
Induction of ROS levels in cells by TMZ, SQ NPs, and SQ-TMZ NPs. (**A**) Microscopic images showing ROS fluorescence intensity (bar = 100 µm). (**B**) Flow cytometry analysis of intracellular ROS levels. Data are presented as mean ± SD (*n* = 3) (mean ± SD, *n* = 3, One-way ANOVA was performed to compare group means, and post hoc multiple comparisons were conducted using Tukey’s HSD test, * vs. SQ-TMZ NPs, *** *p* < 0.001, ** *p* < 0.01).

**Figure 6 ijms-26-04723-f006:**
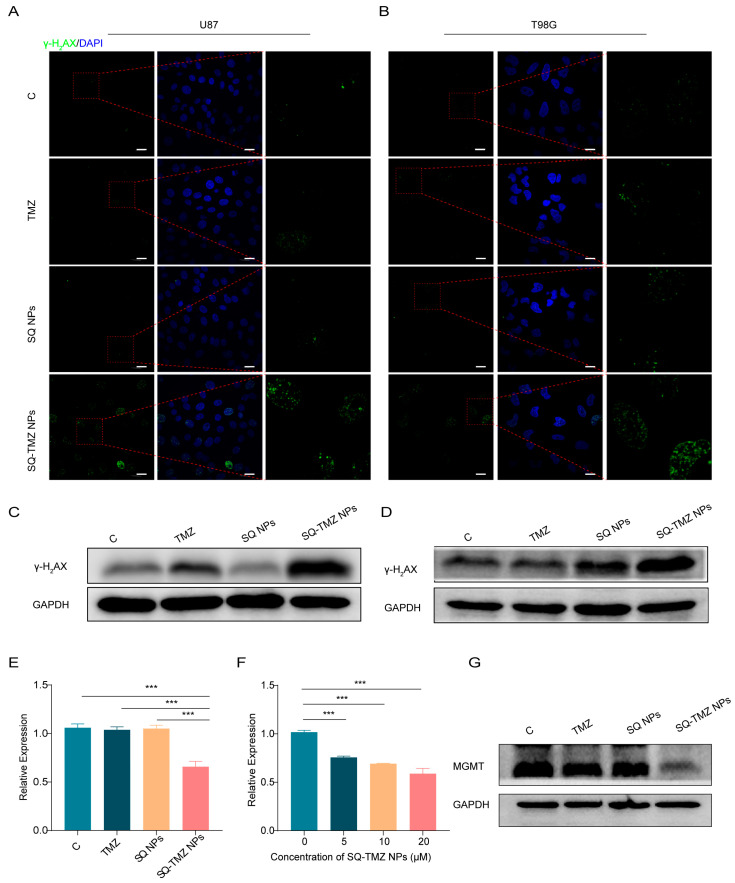
Effects of SQ-TMZ NPs on DNA damage and MGMT transcription and translation levels. (**A**,**B**) Immunofluorescence analysis of γ-H_2_AX protein levels in cells treated with SQ-TMZ NPs to assess DNA damage (bar = 20 µm). (**C**,**D**) Western blot analysis of γ-H_2_AX protein levels in T98G cells treated with SQ-TMZ NPs to evaluate DNA damage. (**E**,**F**) qPCR analysis of MGMT transcription levels in cells treated with SQ-TMZ NPs. (**G**) Western blot analysis of MGMT protein expression levels in T98G cells treated with SQ-TMZ NPs. (mean ± SD, *n* = 3, One-way ANOVA was performed to compare group means, and post hoc multiple comparisons were conducted using Tukey’s HSD test, *** *p* < 0.001.

**Figure 7 ijms-26-04723-f007:**
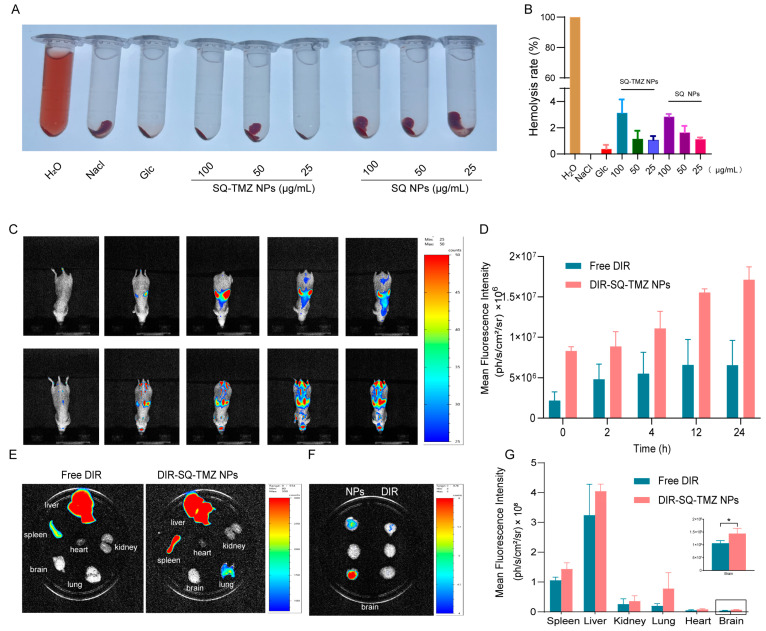
Hemolysis and blood–brain barrier penetration of SQ-TMZ NPs. (**A**) Images of blood samples after centrifugation. (**B**) Hemolysis rate of SQ-TMZ NPs. (**C**,**D**) In vivo imaging of fluorescently labeled SQ-TMZ NPs crossing the blood–brain barrier in animals. (**E**–**G**) Quantitative analysis of tissue distribution in various organs of mice (mean ± SD, *n* = 3, Unpaired *t*-test was performed to compare group means, * *p* < 0.05).

## Data Availability

All data presented are included in the manuscript and Appendix A.

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
