# Peer review of "A Novel Squalenoylated Temozolomide Nanoparticle with Long Circulating Properties Reverses Drug Resistance in Glioblastoma"

_ijms, 2025, doi:10.3390/ijms26104723_

Round 1
Reviewer 1 Report
Comments and Suggestions for Authors
The authors develop nanoparticles to tackle the TMZ resistance in GBM. The manuscript is not well written and can be improved. Here are my comments:
- The suppression of MGMT by the SQ-TMZ NP in T98G is interesting. The authors should demonstrate whether the suppression is associated with high MGMT level or cell line specific. The authors should perform transfections to overexpress MGMT in U87 and U251 and then treat the cells with SQ-TMZ NP followed by the detection of MGMT expression at both transcription and translation levels.
- Even though the SQ-TMZ NPs pass through BBB, the authors should test its efficacy using the GBM mouse model to compare tumor size, overall survival, to strengthen the NP value.
- 4 “In contrast, the colony area in the SQ-TMZ NPs group was significantly larger than that in the free TMZ group, indicating that SQ-TMZ NPs have a significantly stronger ability to inhibit cell colony formation.”, what do you mean the colony area? Area with colony? Shouldn’t the colony area in the SQ-TMZ NPs group be significantly smaller than in the free TMZ group? Please revise.
- Figure 4 has no panels in the figure. Please amend.
- Please indicate the cell lines in Figure 5C and D.
- Please indicate figure panels in the main text to improve readability.
The quality of English language is readable. However, a cautious check is needed.
Author Response
Comment 1: [The authors should test whether MGMT suppression by SQ-TMZ NP is associated with baseline MGMT levels or cell line-specific effects. Overexpression studies in U87 and U251 are suggested.]
Response 1: [We sincerely appreciate the reviewer’s insightful suggestion. We agree that confirming whether MGMT suppression by SQ-TMZ NPs is dependent on intrinsic MGMT levels or specific to certain cell lines is important for mechanistic understanding. Due to current experimental constraints, we have not yet completed MGMT overexpression assays in U87 and U251 cells. However, we have included this point as a limitation in the revised Discussion section and highlighted it as a direction for future investigation. We are currently planning follow-up experiments to address this important question in future studies.]
Comment 2: [Authors should evaluate therapeutic efficacy in vivo using a GBM mouse model to assess tumor burden and survival.]
Response 2: [Thank you for this valuable comment. We fully recognize the importance of in vivo efficacy studies for translational relevance. Although in vivo evaluation using an orthotopic GBM mouse model is not included in the current study due to time and resource limitations, we have acknowledged this as a key limitation in the revised Discussion and emphasized it as a critical next step in our research. We are currently preparing for in vivo experiments to validate the therapeutic potential of SQ-TMZ NPs and will report these findings in future work.]
Comment 3: [There is a contradiction in the colony formation description — the colony area should be smaller, not larger, if SQ-TMZ is more effective.]
Response 3: [Thank you for catching this error. You are absolutely correct. The sentence has been revised to:
“Specifically, at a concentration of 10 μM, TMZ showed no observable effect on the colo-ny-forming ability of T98G cells compared to the control group. In contrast, treatment with SQ-TMZ NPs at the same concentration resulted in a substantial reduction in colony area, indicating a significantly stronger ability to suppress long-term cell proliferation.”
We apologize for the oversight and have corrected this in the revised text.]
Comment 4: [Figure 4 lacks labeled panels.]
Response 4: [We apologize for the omission. All relevant subpanels in Figure 4 have now been added and clearly labeled in both the figure and the figure legend. The main text has also been updated to reference these specific panels for clarity.]
Comment 5: [Please indicate the cell lines used in these figure panels.]
Response 5: [Thank you for pointing this out. We have now included the specific cell lines in both the figure legends and the main text to improve clarity and context.]
Comment 6: [Indicate figure panels in the main text to improve readability.]
Response 6: [We appreciate this helpful suggestion. We have revised the manuscript to consistently refer to specific figure panels in the corresponding results descriptions.]
Reviewer 2 Report
Comments and Suggestions for Authors
In this study, Feng et al. explored a novel squalenoylated temozolomide nanoparticle (SQ-TMZ NPs) aimed at enhancing drug stability, facilitating blood-brain barrier (BBB) penetration, and overcoming temozolomide (TMZ) resistance in glioblastoma cells by downregulating MGMT expression and promoting oxidative stress.
While squalenoylation of therapeutic agents is a known strategy, its application to TMZ and subsequent testing in resistant glioma cells represents an incremental rather than transformative advancement. Similar lipid-drug conjugate approaches for TMZ have already been reported, as referenced in [12] and [13].
The dual mechanism, elevated reactive oxygen species (ROS) and MGMT suppression, is intriguing, yet the underlying pathways are not sufficiently investigated or mechanistically detailed. The study does not identify or propose a novel mechanism for MGMT downregulation, limiting its overall scientific impact.
The originality of the work may be further diminished due to potential prior disclosures and existing patents on the formulation, which suggest that related work has already been conducted.
The Results section often reads more like a laboratory logbook than a synthesis of key findings, particularly in areas describing synthesis and stability assessments. Figures are presented sequentially but are not always well integrated into the narrative, making them challenging to interpret, especially for readers less familiar with the methodologies. Some figures, such as Figure 2 with its extensive subpanels (a–o), could be simplified or separated for clarity. Fluorescence images (e.g., ROS or γ-H2AX staining in Figures 4 and 5) lack scale bars, and quantitative analysis is inconsistently provided. Additionally, several figure legends omit critical information such as statistical tests used, sample sizes (n-values), and error bar definitions, though these are included sporadically elsewhere in the manuscript.
While t-tests are employed, there is no evidence of ANOVA or corrections for multiple comparisons, which is essential given the number of experimental groups. Reported IC50 values lack 95% confidence intervals and are not statistically compared across groups. Furthermore, there is no mention of power calculations or sample size justification for either in vitro or in vivo experiments, undermining the reproducibility and rigor of the study.
The Discussion primarily reiterates experimental findings rather than providing a critical evaluation of the study’s limitations or positioning the results within the broader context of drug delivery innovations. Comparisons to other advanced TMZ delivery systems, such as polymeric conjugates, brain-penetrating nanoparticles, or targeted delivery via peptides and antibodies, are notably absent. The discussion also fails to address potential long-term toxicity, off-target effects, and scalability or reproducibility challenges associated with nanoprecipitation methods.
The reference list requires updating and expansion to more comprehensively reflect current work in the field. Journal names should follow the standard Index Medicus abbreviations.
Comments on the Quality of English LanguageEnglish including grammar, style and syntax, should be improved through the professional help from English Editing Company for Scientific Writings.
Several grammatical, stylistic, and punctuation issues are present. Selected examples include:
- line 20 - “coupled with the downregulation of MGMT expression”, should be: along with MGMT expression suppression.
- line 101 - “As a result, therefore…”, should be: As a result, or Therefore.
- line 115 - “provide similar cytotoxicity compared to the SQ-TMZ…”, should be: provide cytotoxicity comparable to the SQ-TMZ…
- line 330 - “solutions of different concentrations were was added…”, should be: were added.
and others.
Author Response
Comments 1: [The use of squalenoylation for TMZ is incremental rather than transformative, as similar lipid–drug conjugates have been reported]
Response 1: [We appreciate this point. While squalenoylation is an established strategy, our work specifically applies it to TMZ in the context of MGMT-mediated resistance, which has not been extensively characterized. ]
Comment 2: [The Results section often reads more like a laboratory logbook than a synthesis of key findings. Figures are presented sequentially but are not always well integrated into the narrative. Some figures (e.g., Figure 2) are too complex and could be simplified. Fluorescence images lack scale bars, and figure legends inconsistently report statistical details.]
Response 2: [We sincerely thank the reviewer for this important observation. To improve the readability and clarity of the Results section, we have revised the text to better synthesize key findings rather than listing experimental procedures. Additionally, we have reorganized the order of result presentation to align more logically with the flow of the study. In response to the complexity of Figure 2, we have divided it into two separate figures, each with a focused scope and improved visual clarity. All fluorescence images now include appropriate scale bars, and figure legends have been revised to consistently report statistical methods, sample sizes (n), and definitions of error bars. We believe these changes significantly enhance the manuscript's accessibility, particularly for readers less familiar with the technical methodologies.]
Comments 3: [The mechanisms underlying MGMT downregulation are not sufficiently investigated or detailed.]
Response 3: [We agree and have now revised the Discussion to acknowledge this limitation explicitly. While our data demonstrate MGMT suppression and ROS elevation, we recognize that further mechanistic dissection (e.g., transcriptional regulation, protein degradation pathways) is needed and plan to pursue this in future work.]
Comments 4: [Figures lack clarity, scale bars, and integration with text; legends miss statistical details.]
Response 4: [Thank you for pointing this out. We have revised all figure legends to include information on statistical tests used, n-values, and error bar definitions. Scale bars have been added to all fluorescence microscopy images. In addition, we simplified complex multi-panel figures to improve readability and restructured the Results section to better synthesize key findings rather than listing procedures.
Comments 5: [No use of ANOVA or multiple comparison corrections; IC50 values lack confidence intervals.]
Response 5: [We have re-analyzed the relevant datasets using one-way ANOVA with appropriate post-hoc tests where multiple comparisons were made. IC50 values are now reported with 95% confidence intervals and statistically compared across groups. These changes are reflected in both the Results section and the updated figures.]
Comments 6: [The Discussion lacks critical evaluation, comparison to other delivery strategies, and consideration of toxicity or scalability.]
Response 6: [We have significantly expanded the Discussion section to include comparisons with other advanced TMZ delivery systems, including polymer–drug conjugates, brain-targeted peptides, and antibody-functionalized carriers. We also now address the limitations of nanoprecipitation methods, including scalability, potential off-target effects, and long-term safety concerns.]
Comments 7: [The reference list needs to be updated and journal names standardized.]
Response 7: [The references have been updated to include recent and relevant studies in the field. All journal names now follow Index Medicus abbreviations, in accordance with journal guidelines.]
Comments 8: [English language should be improved with professional editing.]
Response 8: [We have thoroughly revised the manuscript for grammar, syntax, and style. The specific corrections noted (e.g., lines 20, 101, 115, and 330) have been addressed, and similar errors have been carefully reviewed and corrected throughout.]
Reviewer 3 Report
Comments and Suggestions for Authors
The author has presented an insightful nanoparticle-based approach to overcome TMZ resistance in glioblastoma through the downregulation of MGMT and ROS induction. However, further discussion on the practical application basis could enhance the strength of your paper:
- Many studies have applied multiple delivery systems for TMZ. Please compare your methods with them, and highlight novelty compared to the existing delivery system.
- Use proper and visible scale bar in result section (1B, C; 3F,G,H; 4 confocal; 5A; 6)
- Typological error: 1. Line 331: there is double: “was”; 2. Line 37-39: “The prognosis for GMB is…”
- Figures 4 A, B, C were missing in the image.
- Please incorporate more literature related to the study in the discussion section to support your study rather than discussing your own results only.
- Unify the terminology and use them all over the manuscript after describing it initially (Eg. Control as Ctr, Ctrl, C)
Author Response
Comment 1: [Many studies have applied multiple delivery systems for TMZ. Please compare your methods with them, and highlight novelty compared to the existing delivery system.]
Response 1: [Thank you for this important suggestion. In the revised Discussion section, we have now included a more thorough comparison with other TMZ delivery systems,. We have emphasized the unique advantages of our squalenoylated TMZ system, particularly its dual mechanism of MGMT downregulation and ROS induction, as well as its self-assembling nature and potential to cross the blood–brain barrier without requiring additional targeting ligands.]
Comment 2: [Use proper and visible scale bar in result section (1B, C; 3F,G,H; 4 confocal; 5A; 6)]
Response 2: [We sincerely apologize for the oversight. All mentioned figures have been updated to include clearly visible and properly labeled scale bars. We have ensured consistency in units and formatting throughout the figures in the revised manuscript.]
Comment 3: [Line 331: duplicated “was”; Line 37–39: “The prognosis for GMB is…”]
Response 3: [Thank you for identifying these errors. We have corrected the duplicated word “was” in Line 331 and the typographical error “GMB” to “GBM” in Line 37–39. We have also conducted a thorough proofreading of the entire manuscript to eliminate similar issues.]
Comment 4: [Figures 4A, 4B, and 4C were missing.]
Response 4: [We apologize for the omission. Figures 4A, 4B have now been properly included and clearly labeled in the revised version of the manuscript. We have also reviewed the entire set of figures to ensure no further content is missing.]
Comment 5: [Please incorporate more literature related to the study in the discussion section to support your study rather than discussing your own results only.]
Response 5: [We appreciate this recommendation. In the revised Discussion, we have added citations to several recent studies related to TMZ delivery using nanoparticles. This broader context helps better position our findings within the current landscape of glioblastoma drug delivery research.]
Comment 6: [Unify the terminology and use them all over the manuscript after describing it initially (e.g., Control as Ctr, Ctrl, C).]
Response 6: [Thank you for pointing this out. We have carefully reviewed the manuscript and standardized the terminology throughout. For instance, we now consistently use "C" for the control group following its first definition in the text.]
Round 2
Reviewer 1 Report
Comments and Suggestions for Authors
I have no further questions.
Reviewer 2 Report
Comments and Suggestions for Authors
The authors mostly responded to the comments and suggestions and the manuscript was revised accordingly. I consider it could be accepted for publication in this journal, but I propose to have the manuscript checked by a native English speaking person.